# TIG-DETR: Enhancing Texture Preservation and Information Interaction for Target Detection

**Zhiyong Liu [1,\*,†], Kehan Wang [1,†], Changming Li [2], Yixuan Wang [1] and Guoqian Luo [1]**

[1] College of Information Science and Technology, Northeast Normal University, Changchun 130024, China; wangkh186@nenu.edu.cn (K.W.); wangyx504@nenu.edu.cn (Y.W.); luogq231@nenu.edu.cn (G.L.)

[2] Engineering Technology Development Center, Changchun Guanghua University, Changchun 130033, China; changming_li0034@163.com

[\*] Correspondence: liuzy452@nenu.edu.cn

[†] These authors contributed equally to this work.

**Abstract:** FPN (Feature Pyramid Network) and transformer-based target detectors are commonly employed in target detection tasks. However, these approaches suffer from design flaws that restrict their performance. To overcome these limitations, we proposed TIG-DETR (Texturized Instance Guidance DETR), a novel target detection model. TIG-DETR comprises a backbone network, TE-FPN (Texture-Enhanced FPN), and an enhanced DETR detector. TE-FPN addresses the issue of texture information loss in FPN by utilizing a bottom-up architecture, Lightweight Feature-wise Attention, and Feature-wise Attention. These components effectively compensate for texture information loss, mitigate the confounding effect of cross-scale fusion, and enhance the final output features. Additionally, we introduced the Instance Based Advanced Guidance Module in the DETR-based detector to tackle the weak detection of larger objects caused by the limitations of window interactions in Shifted Window-based Self-Attention. By incorporating TE-FPN instead of FPN in Faster RCNN and employing ResNet-50 as the backbone network, we observed an improvement of 1.9 AP in average accuracy. By introducing the Instance-Based Advanced Guidance Module, the average accuracy of the DETR-based target detector has been improved by 0.4 AP. TIG-DETR achieves an impressive average accuracy of 44.1% with ResNet-50 as the backbone network.

**Keywords:** object detection; DETR; FPN; transformer; attention mechanism

## 1. Introduction

With the rapid development in the field of deep learning, significant progress has been made in target detection techniques. Many advanced detectors based on CNN and Transformer [1,2] have driven the steady development of the field. Among them, FPN [3], as a simple and effective method, improves detector performance by passing semantic information and implementing a hierarchy of CNN features. In recent years, the Transformer technique has gradually gained attention and has been applied to many target-based detection methods [4–8]. Compared with traditional CNN-based detectors [9,10], these Transformer-based detectors have achieved remarkable results. However, FPN structures and Transformer-based detectors still have design flaws that affect model performance, as described below:

**Loss of texture information in FPNs.** The classical FPN network significantly improves the performance of the detection network through the learning of multi-scale features. Subsequent studies [11–15] have used a similar structure. Several studies have shown that low-level features are helpful for identifying larger targets, and the rich texture information contained in low-level features aids in target localization and accurate bounding box generation. However, features obtained from the backbone network inevitably lose a large amount of texture information during the downsampling process, which may affect the detection network's ability to accurately acquire location information.

**Confounding effect of cross-level feature fusion**. When performing cross-layer feature fusion, the up-sampled feature maps are superimposed with the original feature maps, leading to feature discontinuities and confusion in the fused features, referred to as the blending effect of features [3]. The severity of the blending effect increases as more feature maps are superimposed.

**The limitation of interaction between windows in Shifted Window based Self-Attention**. The global attention mechanism in Transformer-based detection networks introduces a significant computational burden [16]. Although Shifted Window-based Self-Attention reduces the computational complexity, interactions between windows are limited to neighboring windows, potentially weakening the model's perception of larger objects.

To address these issues, we proposed TIG-DETR(Texturized Instance Guidance DETR), which includes a backbone network, a novel FPN (Texture-Enhanced FPN), and a detection module based on DETR. TIG-DETR aims to mitigate the aforementioned flaws and improve model performance.

Our contributions:

1.  We introduced a novel approach to address the issue of missing texture information in FPN by constructing a new bottom-up path that utilizes the low-level feature map in the backbone. Unlike the traditional downsampling process in the backbone network, our method aims to preserve textural information as much as possible by constructing a new bottom-up path. By fusing this texture-rich feature map with the features at the same level as the top-down path in FPN, we obtained a new feature map that contains both rich semantic and texture information. Although previous studies [12,17,18] have utilized bottom-up paths, their convolution-based downsampling still results in significant texture information loss. In contrast, our method effectively retains the texture information in the feature map.

2.  We proposed a new attention module called 'Feature-wise Attention' to mitigate the confounding effects caused by cross-level feature fusion in FPN. This attention module was designed to be lightweight and was applied to augment the final features of the TE-FPN output. By incorporating this attention mechanism, we enhanced the discriminative power of the features.

3.  We introduced the Instance-Based Advanced Guidance Module to overcome the limitations of interactions between windows in Shifted Window-based Self-Attention [16]. This module enhances the model's perception of large object instances by allowing the model to perceive the instances in the image before finer self-attention and enabling information interaction between each window prior to window attention movement. This approach significantly improves the model's ability to capture and understand large object instances.

Overall, our proposed methods contribute to the improvement of texture preservation, feature fusion, and perception ability in the context of model.

In summary, TIG-DETR aims to overcome the design flaws in FPN and Transformer-based detectors by addressing the loss of textural information in FPN, mitigating the blending effect of cross-layer feature fusion, and improving the interaction capability of windows in Shifted Window-based Self-Attention. Ultimately, these improvements enhance the performance of the target detection model.

## 2. Related Work

**FPN**. Before the introduction of FPN, various approaches for feature processing existed, including featurized image pyramids, single feature maps, and pyramidal feature hierarchy. SSD [19] utilized pyramidal feature hierarchy, specifically focusing on hierarchical feature prediction goals, to enable different level features to learn the same semantic information. FPN [3] proposes a method for fusing features of different resolutions by element-wise addition of the feature map from each resolution with the up-sampled low-resolution feature map. This enhancement improves the features at different levels and subsequent models have built upon the FPN foundation. PANet [12] introduced a bottom-up path

enhancement to shorten the information path by utilizing the precise localization signal stored in low-level features, thereby improving the performance of the feature pyramid architecture. EfficientDet [9] borrowed the TopDown-BottomUp concept from PANet and incorporated residual structures in each block to reduce optimization difficulties. Furthermore, the authors recognized that features from different layers possess varying semantic information. Directly summing these features can lead to sub-optimal problems. To address this, the authors introduced a learnable parameter in front of each layer of features to automatically determine their weights. Aug-FPN [13] proposes Soft ROI Selection, which involves pooling ROI features from different levels and fusing them to enhance the performance of the feature pyramid architecture. To mitigate the loss of textural information in high-level feature maps, Aug-FPN incorporates a residual enhancement branch specifically designed to enhance the texture information of these high-level feature maps. In CE-FPN [14], sub-pixel enhancement and attention-guided modules were employed in FPN to fully leverage the rich channel information of each level feature map, while minimizing the loss of channel information during the downscaling process.

**Target detector**. Traditional image target detection can be categorized into two main types: two-stage detectors, with Faster R-CNN [20] being the most representative example, and one-stage detectors such as YOLO [21], YOLO9000 [22], and YOLOV3 [23]. R-CNN [24] demonstrated for the first time the significant improvement in target detection performance by using CNN on the PASCAL VOC dataset [25] compared to HOG-like feature-based systems. Fast R-CNN [26], proposed subsequently, overcomes the time-consuming aspect of R-CNN's SVMs [27] classification by employing ConvNet forward propagation for each region without redundant computation. Fast R-CNN [28] extracts features from the entire input image and passes them through the ROI pooling layer to obtain fixed-size features for subsequent classification and bounding box regression in the fully connected layer. Instead of extracting features for each region separately, Fast R-CNN extracts features once from the entire image, reducing both the processing time and the storage space required for a large number of features. Fast R-CNN [26] adopts selective search to propose RoIs but this approach is slower and has the same running time as the detection network. In contrast, Faster R-CNN [20] introduces a new RPN (region proposal network) that is composed entirely of convolutional networks and efficiently predicts region proposals. The RPN shares the same set of common convolutional layers with the detection network, and the fully convolutional Mask R-CNN [28] optimizes the model by integrating low-level and high-level features to enhance the classification task. YOLO [21] pioneered one-stage target detection, and subsequent one-stage detectors have built upon its improvements. Generally, two-stage detectors achieve higher localization and target detection accuracy, while one-stage detectors offer faster inference speed. However, both types of detectors are influenced by post-processing steps such as compressing redundant prediction results, anchor frame design, and heuristics for assigning target frames to anchor frames [29]. In contrast, DETR [5] achieves an end-to-end target detector by directly predicting without relying on intermediate methods.

**Transformer**. Transformer was initially introduced as a Seq2Seq [30] model designed for machine translation. Subsequent studies have demonstrated that a pre-trained Transformer-based model (PTM) [31] can achieve state-of-the-art performance on various tasks. Consequently, the Transformer has emerged as the preferred architecture for NLP tasks. Besides the NLP domain, the Transformer has gained significant adoption in areas such as computer vision, audio processing, etc. [32]. The Non-local Network [33] was the first to employ the self-attentive mechanism in the field of computer vision, achieving successful target detection. Several frameworks have been proposed in recent years [18,34,35] to enhance the Transformer and optimize it from various perspectives. Visual transformers [36] incorporate the Transformer into a CNN, enhancing the CNN network by allocating semantic information of the input image to different channels and closely correlating them through encoder blocks (referred to as VT blocks). VT blocks are employed as an alternative to partial convolution to improve the semantic modeling capacity of

CNN networks. SWIN-T [18] introduced Shifted Window based Self-Attention, significantly reducing the computational complexity of the transformer when processing images. Funnel Transformer [37] employs a funnel-like encoder architecture that incorporates pooling along the sequence dimension to progressively decrease the length of the hidden sequence and then employs upsampling for reconstruction, effectively reducing FLOP and memory consumption. When employing the transformer in the computer vision (CV) domain, the feature space resolution is constrained, and the network encounters challenges in convergence during training. To address these issues, Zhu et al. [4] introduced Deformable DETR, which accelerates model convergence by directing the attention module to concentrate on a subset of key sampling points surrounding the reference.

**Attention mechanism**. Attention plays a crucial role in human perception of external information, as humans selectively concentrate on the most salient parts when processing information in a scene to enhance the capture of relevant information [38]. RAM [39] integrates deep neural networks with an attention mechanism, enabling end-to-end updating of the entire network by iteratively predicting significant regions. This marks the first implementation of an attention mechanism in CNN networks. Numerous subsequent works have adopted comparable attention strategies. STN [40] predicts the spatial transformation by incorporating a sub-network that identifies significant regions in the input. SE-Net [41] introduces a compact module that enhances inter-channel relationships by utilizing global average pooling to compute attention across channels. GSoP-Net [42] addresses the limitation of using global average pooling alone in SENet for collecting contextual information, which restricts the modeling capacity of the attention mechanism. To overcome this, GSoP-Net proposes the global second-order pooling (GSoP) block to capture higher-order statistics while incorporating global contextual information. CBAM [43] incorporates global maximum pooling in addition to global average pooling, boosting the attention mechanism's response to maximum gradient feedback. Furthermore, the combination of spatial attention and channel attention demonstrates superior performance compared to channel attention alone. Adding spatial attention to channel attention verifies that using both is better than using channel attention alone. In our Feature-wise Attention, we introduced soft pooling [44] as novel contextual information to provide distinct gradient feedback for individual features. This approach assigns different attention weights to different features, thereby enhancing the preservation of textural information in the image instances.

## 3. Materials and Methods

As shown in Figure 1, we propose Texturized Instance Guidance DETR (TIG-DETR) architecture comprises a backbone, an FPN network, and a DETR-based detector. In order to enhance the model's localization capability, we introduced a bottom-up path in the FPN that retains the texture information of the feature maps and combine it with the pyramidal features produced in the top-down path of the FPN. This fusion results in a feature map that contains both abundant semantic and texture information. The SRS (Soft RoI Selection) module was employed to integrate the features produced at each level of the feature pyramid, and Feature-wise Attention was utilized to enhance the features of the resulting output feature map. Within the DETR-based detection model, we employed local self-attention to improve the recognition performance of small object instances and facilitate faster model convergence. Additionally, we introduced a module to address the limitations of local attention in perceiving large object instances, thereby enhancing the model's capability to detect small object instances without compromising its performance in detecting large object instances.

**Figure 1.** TIG-DETR comprises a backbone network, a new pyramidal structure known as Texture-Enhanced FPN (TE-FPN), and an enhanced DETR detector.

### 3.1. TE-FPN

The top-down propagation of robust semantic information by FPN enhances the model's ability to accurately classify features at all levels of the pyramid. The accurate localization of instances in the model relies on their high response to instance parts or edges, whereas the bottom-up path approach effectively propagates robust texture information, thereby enhancing the model's ability to localize features at all levels of the feature pyramid. In this paper, we proposed a new pyramid structure Texture-Enhanced FPN (TE-FPN), which contains a bottom-up path leading from the low level of the backbone network, so that the fused feature map has both strong semantic and textural information. Additionally, we introduced a novel channel attention mechanism to the Soft RoI Selection process, aiming to further enhance the fused features.

**Enhancing textural information with a bottom-up architecture.** FPN acquires features from the backbone and a large amount of texture information is inevitably lost when the backbone is downsampled, a situation that may affect the accuracy of the detection network in obtaining information about the location of instances in the image. To address this limitation, we incorporated an 'enhancing textural information with a bottom-up architecture' (ETA) into FPN, aiming to enhance the texture information in the feature map at each level. Following the definition of FPN [3], feature layers of the same size were generated in each network phase, and different feature layers correspond to different phases of the network. As shown in Figure 2, the Resnet-50 [45] serves as the backbone network and $\{P2, P3, P4, P5\}$ represent the feature layers generated by the FPN. $\{C2, C3, C4, C5\}$ represent the feature layers at different stages in the backbone network and $\{D2, D3, D4, D5\}$ represent the feature layers of $\{C2, C3, C4, C5\}$ after dimensionality reduction using convolution. From $C2$ to $C5$, $P2$ to $P5$, and $D2$ to $D5$ spatial sizes were gradually downsampled with a downsampling factor of 2. $\{N2, N3, N4, N5\}$ represent the feature maps newly generated by the bottom-up path, corresponding to $\{C2, C3, C4, C5\}$.

Specifically, the first step was to reduce $N2$ to $C2$ using convolution with a channel dimension of 256. This channel dimension aligns with the feature map in FPN and enables effective fusion between the two feature maps. Subsequently, the downsampled feature map was further downsampled with sampling coefficients of 2, 4, and 8 to generate $\{N3, N4, N5\}$, preserving more texture information compared to conventional convolutional downsampling. The process is described as follows:

$$N_i = pool_{2^{i-2}}(N_2), \tag{1}$$

where $pool_\gamma$ denotes the downsampling of the sampling factor of $\gamma$.

Finally, depicted in Figure 3a, we up-sampled Pi using a sampling factor of 2 and merged the up-sampled feature map with $N_{i+1}$ and $D_{i+1}$, which have the same size, to generate $P_{i+1}$. Notably, $P5$ was obtained by merging only $D5$ and $N5$. The resulting fused feature map contains a combination of rich semantic and textural information. To mitigate the confounding effect after feature fusion, we employed a Lightweight Feature-wise Attention (LFA) module, as shown in Figure 3b. In this module, we implemented a lightweight attention mechanism using FC layers instead of the more complex shared MLP

layers and combined the output feature vectors through element-wise summation with a sigmoid function. The process can be summarized as follows:

$$LFA(F) = \sigma(fc1(Avgpool(x)) + fc2(Maxpool(x)),$$
$$+ fc3(Softpool(x))) \qquad (2)$$

where *LFA* denotes the lightweight channel attention function, $\sigma$ denotes the sigmoid function, *Avgpool, Maxpool, Softpool* denotes the global average pooling, global maximum pooling, and global soft pooling. Respectively, Lightweight Feature-wise Attention was used to mitigate the confounding effect after feature fusion, rather than enhancing the features themselves.

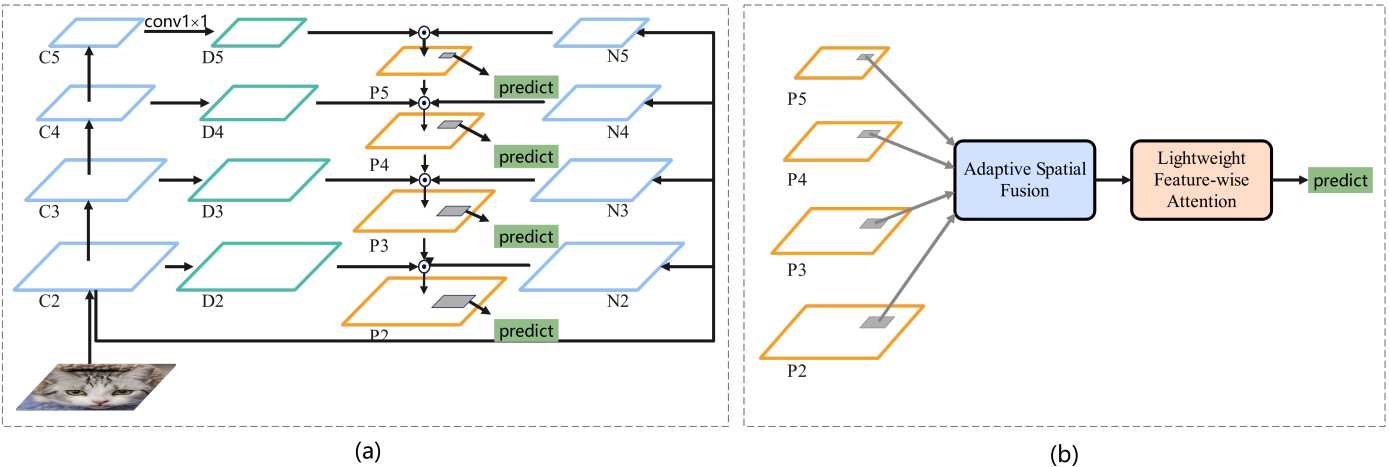

**Figure 2.** Overall structure diagram of TE-FPN. The proposed approach introduces an architecture called enhancing textural information with a bottom-up architecture (ETA), which allows the input of low-level features to each level of the feature hierarchy. The lightweight Feature-wise Attention (LFA) was employed to extract channel weights using the channel attention module, which were then used to generate the final integrated features. The Feature-wise Attention (FWA) leverages multiple contextual information to acquire channel weights and enhance the features of the final output feature map. (**a**) Schematic diagram illustrating the architecture of enhancing textural information with a bottom-up approach. (**b**) Schematic diagram illustrating the enhanced features using Feature-wise Attention.

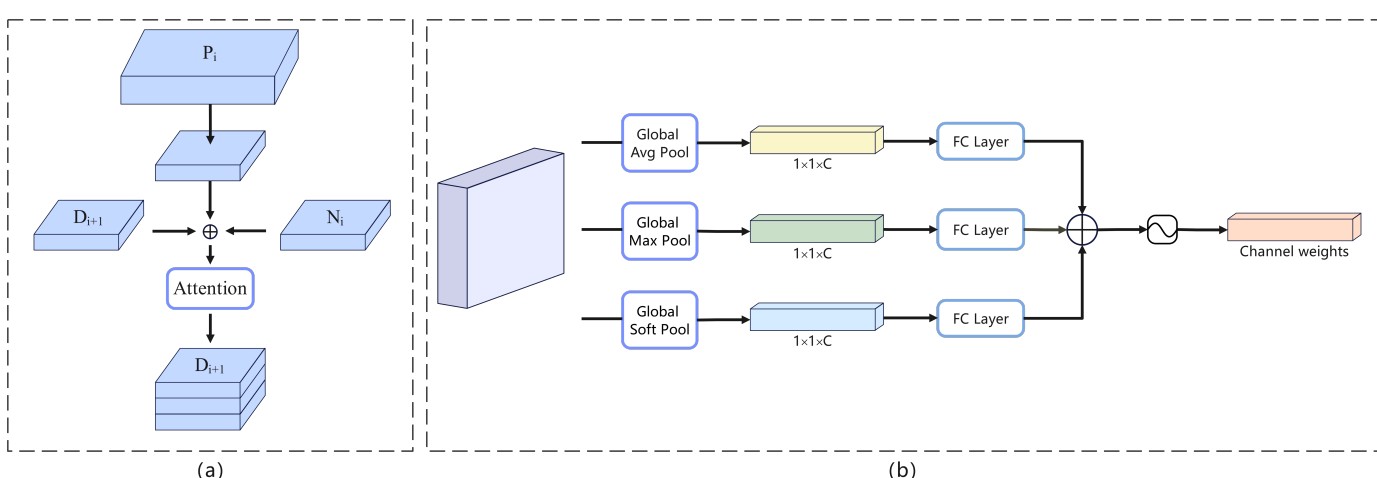

**Figure 3.** (**a**) The schematic diagram illustrates our proposed bottom-up architecture for the feature fusion module, enhancing texture information. (**b**) The diagram illustrates the schematic of the Lightweight Feature-wise Attention.

**Feature-wise Attention**. During the target detection task, the detector heavily relies on the edge and textural information of the instances in the image to accurately delineate the instances. However, the multi-scale feature fusion can introduce a blending effect, leading to discontinuity in the fused feature map features. This can result in the detector acquiring incorrect edge and texture information of the instances, ultimately affecting the accuracy of instance localization and detection tasks. To mitigate the influence of the blending effect on the model, we introduced a novel attention module called 'Feature-wise Attention' in its lightweight version, specifically designed to address the impact of the blending effect on model performance. In this work, we replaced the ROI module with the SRS for feature fusion across different scales in the feature pyramid. Additionally, we incorporated the standard Feature-wise Attention to enhance the features in the final output.

The Feature-wise Attention (FWA) is illustrated in Figure 4. In this mechanism, we first utilized global average pooling, global soft pooling, and global maximum pooling to obtain three different spatial contexts. These contexts captured various aspects of the feature map. Next, each context was processed through an MLP layer with shared parameters. Finally, the resulting feature vectors were combined using element-wise summation followed by a sigmoid function. The channel attention in FWA focused on identifying significant features within the graph. Global average pooling provides feedback for every pixel point on the feature map, while global maximum pooling focuses on gradients by considering only the areas with the highest response. On the other hand, soft pooling [44] produces diverse gradient feedback for different pixel points during gradient backpropagation. Global average pooling tends to capture overall image features, global maximum pooling emphasizes instance edge information, and global soft pooling captures the overall texture information of the instance. By incorporating the global soft pooling contextual information and assigning higher weights to each pixel point of the instance, we enhanced the texture information of the instance in the image. The Feature-wise Attention mechanism can be summarized as follows:

$$
\begin{aligned}
FWA(F) = \sigma(mlp(Avgpool(x)) + mlp(Maxpool(x)), \\
+ mlp(Softpool(x)))
\end{aligned}
\tag{3}
$$

where *FWA* denotes the attention mechanism and $\sigma$ denotes the sigmoid function. By adding *FWA*, the textural information of the instances in the image is enhanced to obtain better localization.

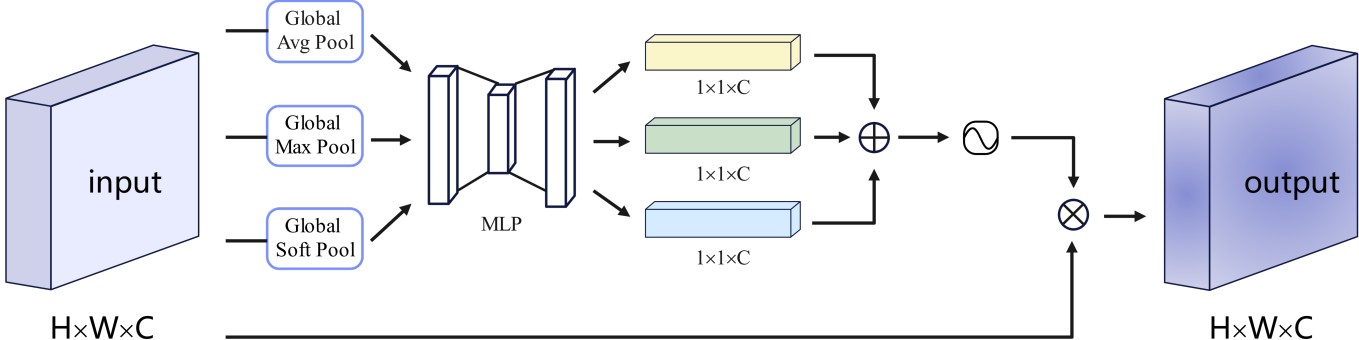

**Figure 4.** Schematic representation of Feature-wise Attention.

### 3.2. Instance Based Advanced Guidance Module

The Transformer used in our TIG-DETR detector follows the structure of Shifted Window-based Self-Attention in SWIN-T. It replaces the multi-headed self-attentive module in DETR with W-MSA and SW-MSA in an alternating manner. The main goal was to reduce the computational complexity of the Transformer part in DETR, enhance the detection performance of small object instances, and speed up the model convergence. However, the limitation of interaction between windows in Shifted Window-based Self-Attention

affects the detection performance of large object instances. To address this, we introduced a new module called Instance-Based Advanced Guidance Module (IAM) before the encoder. This module allows the model to perceive the instances in the image before performing local self-attention, compensating for the degraded detection performance of large object instances caused by the window interaction limitation.

Specifically, as shown in Figure 5, the images from different stages in the backbone underwent a scale-invariant downsampling process to achieve a consistent channel dimension. They were then resized to the final output size. Afterwards, they were fused with the output image at multiple scales, enabling the fused feature map to combine information from different scales and enhance the texture information of the image. After undergoing an LFA, the fused feature map, originally of size $w \times h \times C$, was divided into $M^2$ patches. In Figure 5, we used M = 2 as an example. These patches were fused together through concatenation, resulting in a patch of size $(h/M) \times (w/M) \times CM^2$. The different colors within the fused patches represent the channel information of the patches at different locations. Each pixel within the fused patch contains positional information from the patches at different locations, allowing the model to extract global features, contextual relationships, and better perceive objects of varying sizes. The fused patch was then passed through the multi-headed self-attentive module and the output patch was used to reconstruct the original feature map. We believe that this method is advantageous for improving Shifted Window-based Self-Attention, as it enables the feature map to capture instances before window attention. Despite a slight increase in computational complexity compared to Shifted Window-based Self-Attention, we have successfully implemented this approach. The details are as follows:

$$\Omega(MSA) = 4hwC^2 + 2(hw)^2C \tag{4}$$

$$\Omega(W - MSA) = 4hwC^2 + 2N^2hwC \tag{5}$$

$$\Omega(IAM) = 4hwC^2M^2 + 2(h/M)^2(w/M)^2C. \tag{6}$$

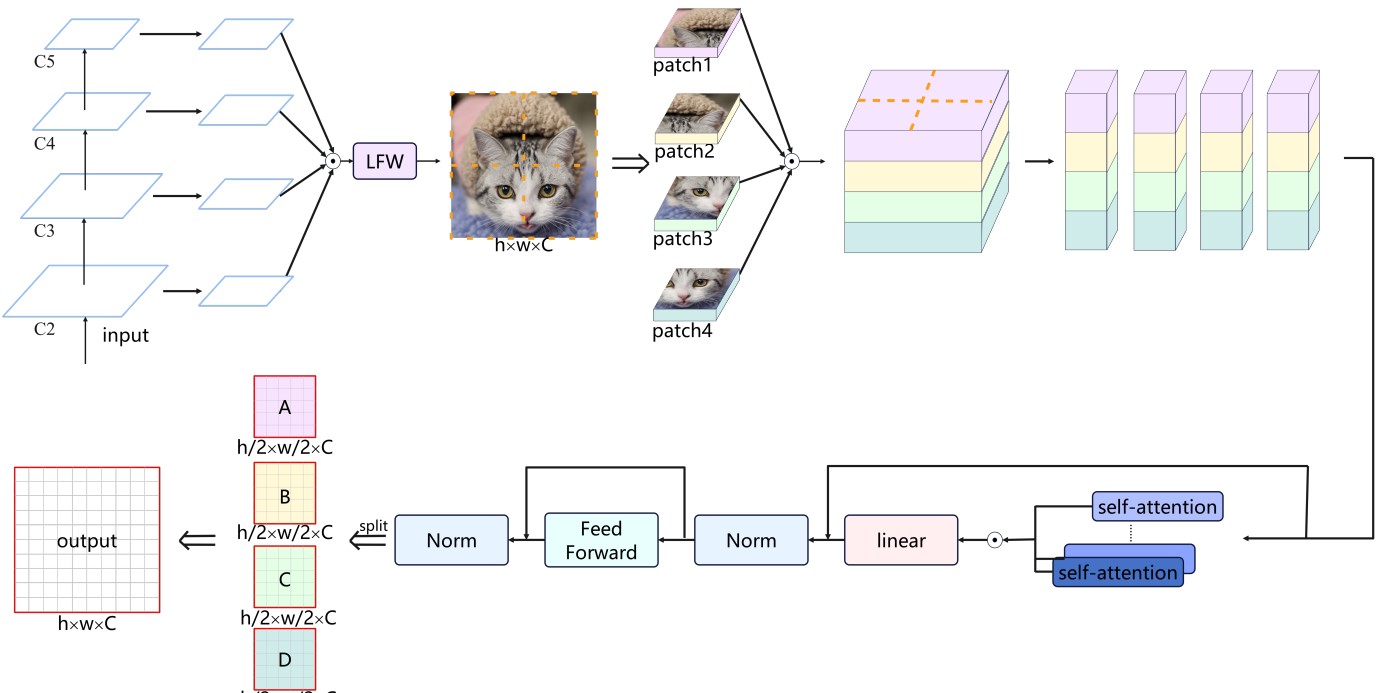

**Figure 5.** Schematic diagram of Instance-Based Advanced Guidance Module.

In the equation, *N* represents the edge length of the shifted window. It is observed that the computational complexity of the global MSA increases quadratically with *hw*.

When $N$ remains constant, the computational complexity of W-MSA becomes linear. In our IAM, with $M$ fixed, the computational complexity of $2(h/M)^2(w/M)^2C$ in this part is much smaller than $2(hw)^2C$. However, $4hwC^2M^2$ for $4hwC^2$ is only a linear increase, much smaller than the difference between the computation of $2(h/M)^2(w/M)^2C$ and $2(hw)^2C$. Consequently, the computational complexity of IAM is still significantly lower compared to the global MSA.

By performing the mentioned operations, we enhanced the textural information of the image and establish associations among individual pixel points within the image prior to applying movable window attention to the entire image. This allows the model to perceive instances in the image before the finer self-attention operation, thereby improving its ability to detect large object instances. A feedforward network (FFN) was employed after the IAM module.

We presented the generalized Instance-Based Advanced Guidance Module, which can be applied to various backbone networks without the need for FPNs. In the case of a backbone with FPNs, we utilized FPNs to replace the multiscale fusion component of the model.

## 4. Results

We conducted target detection experiments using TIG-DETR on the COCO. We compared TIG-DETR and its individual components with other techniques. Additionally, we conducted instance segmentation comparison experiments between TE-FPN and other techniques on the Cityscapes.

### 4.1. COCO and Evaluation Metrics

We compared our techniques with others on the challenging COCO [46], which comprises over 1.5 million instances, including 80 target classes such as pedestrians, cars, elephants, etc., and 91 material classes such as grass, walls, and sky. Each image is accompanied by five descriptive sentences and contains 250,000 pedestrians with key point annotations. For our experiments, we used 115,000 images from the train2017 subset for training, 5000 images for validation (val2017), and 20,000 images for the test-dev set (labels for test-dev are not publicly released). We trained our model on the train2017 subset and evaluated its performance on the val2017 set for the ablation study, as well as on the test-dev set for final evaluation. All reported results adhere to the average accuracy (AP) metrics of COCO. AP represents the average accuracy by considering IoU thresholds ranging from 0.5 to 1.0 with a step size of 0.05. $AP_{50}$ represents the average accuracy at an IoU threshold of 0.5, and $AP_{75}$ represents the average accuracy at an IoU threshold of 0.75, which is a more stringent criterion. $AP_S$ measures the average precision for detecting small targets with a pixel area less than $32^2$. $AP_M$ measures the average precision for detecting medium targets with a pixel area between $32^2$ and $96^2$. $AP_L$ measures the average precision for detecting large targets with a pixel area greater than $96^2$.

### 4.2. Implementation Details

By default, we trained the TIG-DETR model using the AdamW [47] optimizer on 8 GPUs for 50 epochs. The initial learning rate was set to $2 \times 10^{-4}$ and, after the 40th epoch, the learning rate was reduced by a factor of 0.1. For the TE-FPN model, we trained it on 8 GPUs for 15 epochs. During training, we extracted 16 images from one image to generate training samples. The initial learning rate was set to 0.02 and it was reduced by a factor of 0.1 after the 10th and 14th epochs, respectively.

### 4.3. Main Results

We evaluated TIG-DETR and its components on the COCO test development set and compared them with advanced two-stage detectors. The final results are presented in Table 1. We compared TIG-DETR with DETR and, when using only ResNet-50 as the backbone without FPN, our model achieved an AP score that was only 0.6 lower than DETR.

TIG-DETR, which utilizes a local attention mechanism, significantly improves the model's convergence speed, which is only one-tenth of DETR. By introducing IAM to alleviate the limitations of local attention in detecting large object instances, the model's accuracy in detecting large objects decreases slightly and, after adding TE-FPN, the AP reaches 45.9. This demonstrates that IAM has a significant impact on improving the performance of local attention in detecting large object instances. We further adjusted IAM by removing the multiscale fusion and incorporating TE-FPN, resulting in a final AP of 44.1 for TIG-DETR. We also applied IAM to other DETR detectors based on the local self-attention mechanism [4], as shown in Table 1, demonstrating its effectiveness across different models. Notably, IAM shows remarkable improvements in detecting large object instances, highlighting its robustness and versatility. Visualization results are presented in Figure 6.

**Table 1.** Comparison with baseline and state-of-the-art COCO test development methods.

| Method | Backbone | Schedule | AP | $AP_{50}$ | $AP_{75}$ | $AP_S$ | $AP_M$ | $AP_L$ |
|---|---|---|---|---|---|---|---|---|
| Faster R-CNN ∗ | ResNet-50-FPN | ×1 | 36.4 | 58.1 | 39.1 | 21.3 | 40.5 | 44.6 |
| Faster R-CNN ∗ | ResNet-101-FPN | ×1 | 38.6 | 60 | 42.1 | 22.2 | 42.5 | 47.1 |
| Faster R-CNN ∗ | ResNet-101-FPN | ×2 | 39.4 | 61.1 | 43.2 | 22.6 | 42.7 | 50.1 |
| Faster R-CNN ∗ | ResNext-101-32x4d-FPN | ×1 | 40.3 | 62.6 | 43.6 | 24.5 | 42.9 | 49.9 |
| Faster R-CNN ∗ | ResNext-101-64x4d-FPN | ×1 | 41.7 | **64.9** | 44.4 | 24.7 | 45.8 | 51.3 |
| Mask R-CNN ∗ | ResNet-50-FPN | ×1 | 37.1 | 58.9 | 40.3 | 22.3 | 40.5 | 45.5 |
| Mask R-CNN ∗ | ResNet-101-FPN | ×1 | 39.1 | 61.2 | 42.2 | 22.8 | 42.3 | 49.2 |
| Mask R-CNN ∗ | ResNet-101-FPN | ×2 | 40 | 61.8 | 43.7 | 22.7 | 43.4 | 52.1 |
| RetinaNet ∗ | ResNet-50-FPN | ×1 | 35.8 | 55.7 | 38.7 | 19.4 | 39.7 | 44.9 |
| RetinaNet ∗ | MobileNet-v2-FPN | ×1 | 32.9 | 52.1 | 34.9 | 17.9 | 34.8 | 42.6 |
| DETR | ResNet-50 | ×1 | 42 | 62.4 | 44.2 | 20.5 | 45.8 | **61.1** |
| Deformable DETR | ResNet-50 | ×1 | 43.8 | 62.6 | 47.7 | 26.4 | 47.1 | 58 |
| Deformable DETR+IAM | ResNet-50 | ×1 | **44.3** | 62.9 | 48.3 | **26.3** | **47.6** | 60.3 |
| Faster R-CNN * (ours) | ResNet-50-TE-FPN | ×1 | 38.4 | 61 | 41.9 | 23.1 | 41.7 | 47.5 |
| Faster R-CNN (ours) | ResNet-101-TE-FPN | ×1 | 40.2 | 62.6 | 43.6 | 23.5 | 43.5 | 50.9 |
| Faster R-CNN (ours) | ResNet-101-TE-FPN | ×2 | 41.1 | 63.4 | 44.3 | 23.6 | 44.1 | 52.7 |
| Faster R-CNN (ours) | ResNext-101-32x4d-TE-FPN | ×1 | 41.5 | 63.8 | 45.1 | 24.8 | 45.1 | 52.3 |
| Faster R-CNN (ours) | ResNext-101-64x4d-TE-FPN | ×1 | 42.7 | 65.4 | 46 | 25.9 | 45.9 | 53.5 |
| Mask R-CNN (ours) | ResNet-50-TE-FPN | ×1 | 38.9 | 61.1 | 42.4 | 23.2 | 42.2 | 49 |
| Mask R-CNN (ours) | ResNet-101-TE-FPN | ×1 | 40.4 | 63 | 44.2 | 23.7 | 43.3 | 51.4 |
| Mask R-CNN (ours) | ResNet-101-TE-FPN | ×2 | 41.5 | 63.6 | 45.7 | 24.1 | 44.2 | 53.2 |
| RetinaNet (ours) | ResNet-50-TE-FPN | ×1 | 36.9 | 57.9 | 39.6 | 20.8 | 40.1 | 46.4 |
| RetinaNet (ours) | MobileNet-v2-TE-FPN | ×1 | 33.9 | 53.7 | 35.8 | 18.5 | 35.7 | 43.9 |
| TIG-DETR | ResNet-50 | ×1 | 43.1 | 62.1 | 46.2 | 24.7 | 46.8 | 60.5 |
| TIG-DETR | ResNet-50-TE-FPN | ×1 | **44.1** | 62.8 | 48.4 | 25.6 | 47.9 | 62.4 |

* We compared our TIG-DERE and TE-FPN with some advanced target detection networks and FPN networks in the COCO dataset for our experiments. The symbol '∗' means our re-implemented results throughdetection. The bolded part of the font indicates the largest indicator in the comparison experiment.

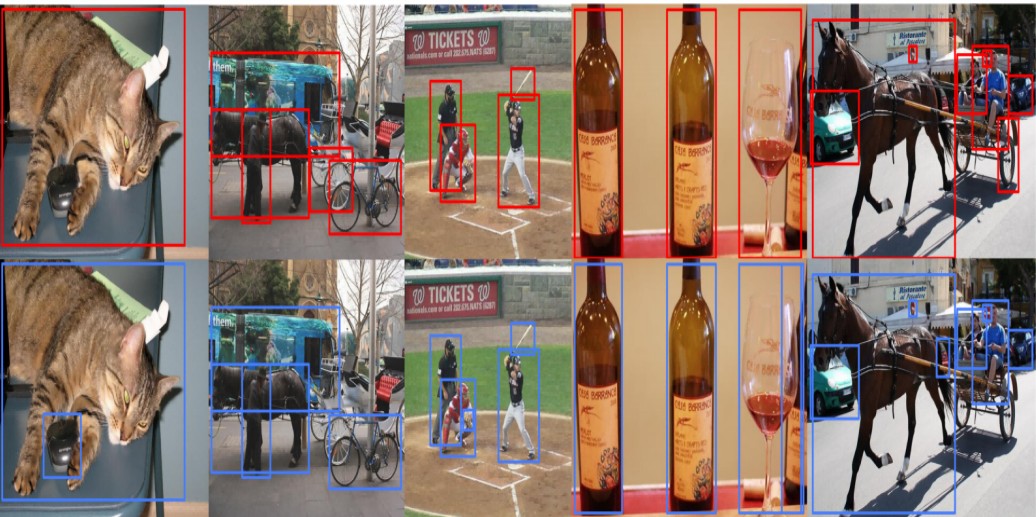

**Figure 6.** TIG-DETR vs. DETR. red bounding box shows the detection result of DETR, while blue bounding box shows the detection result of TIG-DETR.

By replacing FPN with TE-FPN, we achieved an AP of 38.4 for Faster R-CNN using ResNet-50 as the backbone, which is 2.0 points higher than Faster R-CNN based on ResNet-50-FPN. Moreover, TE-FPN also performs well with more powerful backbone networks. For instance, when using ResNext-101-32x4d, our approach improves the AP by an additional 1.2 points. Table 1 demonstrates the varying degrees of performance improvement achieved by TE-FPN across different backbones, detectors, and tasks, highlighting its robustness and generalization capability. Visualization results are presented in Figure 7.

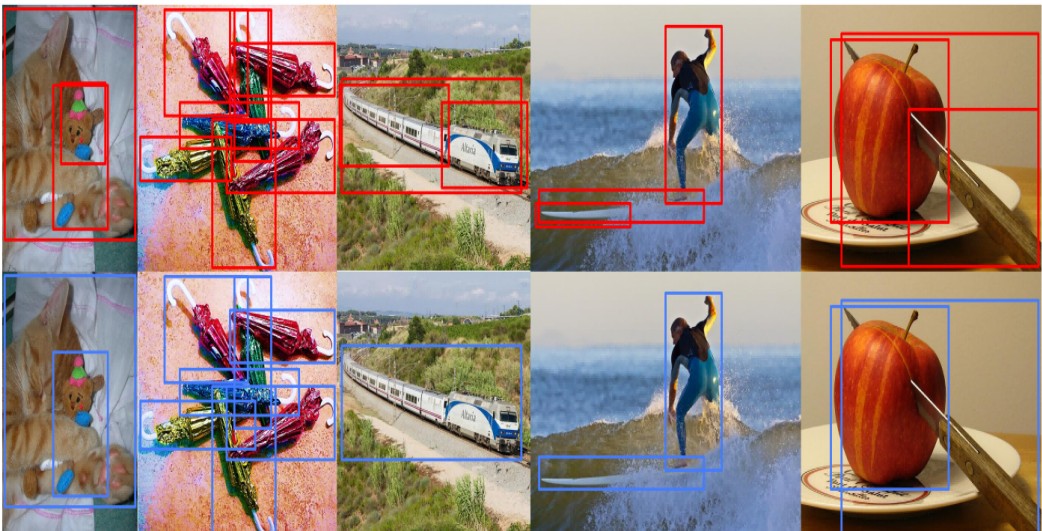

**Figure 7.** TE-FPN vs FPN. red bounding box shows the detection result of FPN, while blue bounding box shows the detection result of TE-FPN.

### 4.4. Ablation Study

In this section, we conducted ablation experiments to analyze the impact of each component in our proposed TIG-DETR and TE-FPN modules.

#### 4.4.1. TIG-DETR

To analyze the significance of each component in TIG-DETR, we systematically incorporated TE-FPN and IAM into the model to assess the influence of each component on the model's performance. The results of all the experiments are presented in Table 2.

**Table 2.** TIG-DETR ablation experiments on COCO.

| IAM | S-IAM | TE-FPN | AP | AP$_{50}$ | AP$_{75}$ | AP$_S$ | AP$_M$ | AP$_L$ |
|---|---|---|---|---|---|---|---|---|
| | | | 40.3 | 60.5 | 42.9 | 22.2 | 44.5 | 57.4 |
| √ | | | 43.1 | 62.1 | 46.2 | 24.7 | 46.8 | 60.5 |
| | √ | | 40.7 | 60.6 | 44.1 | 22.1 | 44.4 | 59.1 |
| | | √ | 43.7 | 62.4 | 47.6 | 26.7 | 47.6 | 60.7 |
| | √ | √ | **44.1** | **62.8** | **48.4** | **26.6** | **47.9** | **62.4** |

IAM denotes Instance-Based Advanced Guidance Module, S-IAM denotes Instance-Based Advanced Guidance Module after removal of multiscale fusion, TE-FPN denotes Texture-Enhanced FPN. The bolded part of the font indicates the largest indicator in the comparison experiment.

According to Table 2, the inclusion of the Instance-Based Advanced Guidance Module enhances the model's accuracy by 2.8 APs. Additionally, the introduction of multiscale fusion leads to improved detection performance for instances of various sizes, particularly for large object instances with a notable improvement of 3.1 APs.

The inclusion of the Instance-Based Advanced Guidance Module, which excludes multiscale fusion, solely enhances the detection performance of large object instances by 1.7 APs. The overall improvement for the model is 0.4 APs.

The introduction of Texture-Enhanced FPN results in a remarkable improvement of 3.4 APs in the model, highlighting the substantial performance enhancement brought by Texture-Enhanced FPN to TIG-DETR.

### 4.4.2. TE-FPN

In order to analyze the significance of each component in TE-FPN, we incrementally incorporated the bottom-up path, LFA, and FWA into the model to evaluate the effectiveness of each component. The results also demonstrate the synergistic effect of combining different components, highlighting their complementary nature. The baseline model for this ablation study was a Faster R-CNN with Resnet-50 as the backbone. The detailed results are presented in Table 3.

**Table 3.** TE-FPN ablation experiments on COCO.

| SRS | ETA | LFA | SRS+FWA | AP | $AP_{50}$ | $AP_{75}$ | $AP_S$ | $AP_M$ | $AP_L$ |
|-----|-----|-----|---------|------|------|------|------|------|------|
|  |  |  |  | 36.2 | 56.1 | 38.6 | 20.0 | 39.6 | 47.5 |
|  | √ |  |  | 36.8 | 59.1 | 39.8 | 20.7 | 40.2 | 48.3 |
|  |  | √ |  | 37.0 | 56.7 | 39.9 | 20.8 | 40.3 | 48.1 |
|  |  | √ |  | 36.8 | 56.5 | 39.3 | 20.6 | 40.2 | 48.0 |
|  |  |  | √ | 37.5 | 57.4 | 40.1 | 21.5 | 40.7 | 49.0 |
|  | √ | √ |  | 37.5 | 57.5 | 40.2 | 21.4 | 41.0 | 49.4 |
|  | √ |  | √ | 37.6 | 58.0 | 40.1 | 21.5 | 41.3 | 49.6 |
|  |  | √ | √ | 37.8 | 57.9 | 40.4 | 21.6 | 41.2 | 49.8 |
|  | √ | √ | √ | **38.2** | **58.8** | **40.9** | **21.9** | **42.3** | **50.4** |

We used Resnet-50+FPN+Faster-R-CNN as our baseline method and gradually added enhancing texture information with a bottom-up architecture (ETA), Lightweight Feature-wise Attention (LFA), Feature-wise Attention (FWA). SRS is an abbreviation for the Soft RoI Selection method mentioned in the paper. The bolded part of the font indicates the largest indicator in the comparison experiment.

According to Table 3, the incorporation of bottom-up paths into TE-FPN improves the baseline approach by 0.8 AP. This demonstrates the significant impact of enhancing texture information in the feature map on enhancing model performance.

By incorporating LFA into the baseline method, the AP improves from 36.2 to 36.8. This indicates that the lightweight Feature-wise Attention has a significant impact on reducing the confounding effect caused by cross-scale feature fusion.

By replacing the ROI of the baseline method with SRS, the AP improves by 0.7. Furthermore, the addition of FWA to the model results in an additional improvement of 0.6 AP. This highlights the significant enhancement effect of FWA on the features.

### 4.5. Cityscapes

We conducted additional experiments on TE-FPN using the Cityscapes [48] to assess its effectiveness in performing instance segmentation tasks. The Cityscapes dataset consists of street scenes captured by in-vehicle cameras, containing numerous overlapping and blurred instances. We utilized 2.9K images for training, 0.5K images for validation, and 1.5K images with fine annotations for testing. Additionally, 20K images with coarse annotations were included but excluded from training. We present the results on the validation and secret test subsets, evaluating the model performance based on AP and $AP_{50}$ metrics.

The instance segmentation task focused on eight object classes: person, rider, car, truck, bus, train, motorcycle, and bicycle. We trained the model using eight GPUs, with eight images randomly sampled from each training image. The initial learning rate was set to 0.01 and it was reduced to 0.001 after 18K iterations. The test performance results are presented in Table 4.

We utilized Mask R-CNN with ResNet-50 as the baseline model and replaced the FPN with TE-FPN. By pre-training our TE-FPN model on COCO, we achieved a performance improvement of 2.7 APs over Mask R-CNN for "fine-only" data. As shown in Table 4, TE-FPN

consistently demonstrates notable results for the instance segmentation task, highlighting the model's strong generalization capability and its effectiveness across different tasks.

**Table 4.** The effectiveness of TE-FPN on Cityscapes.

| Method | AP [val] | AP | AP$_{50}$ | Person | Rider | Car | Truck | Bus | Train | Motorcycle | Bicycle |
|---|---|---|---|---|---|---|---|---|---|---|---|
| Mask R-CNN [fine-only] | 31.5 | 26.2 | 49.9 | 30.5 | 23.7 | 46.9 | 22.8 | 32.2 | 18.6 | 19.1 | 16.0 |
| Mask R-CNN [COCO] | 36.4 | 32.0 | 58.1 | 34.8 | 27.0 | 49.1 | 30.1 | 40.9 | 30.9 | 24.1 | 18.7 |
| TE-FPN [fine-only] | 34.2 | 29.5 | 54.8 | 34.0 | 27.8 | 52.7 | 25.6 | 35.2 | 23.0 | 21.1 | 19.1 |
| TE-FPN [COCO] | **39.6** | **34.9** | **61.2** | **39.1** | **31.1** | **54.3** | **31.5** | **43.9** | **31.1** | **26.2** | **22.4** |

Results on Cityscapes val subset, denoted as AP [val], and on Cityscapes test subset, denoted as AP. The bolded part of the font indicates the largest indicator in the comparison experiment. The bolded part of the font indicates the largest indicator in the comparison experiment.

*4.6. Discusses*

We conducted measurements of the training and testing time for TIG-DETR and its component TE-FPN. Specifically, we compared our model with the baseline model using ResNet50 as the backbone on the same batch size COCO. In the comparison of TE-FPN, the training time for Faster RCNN with TE-FPN was approximately 1.05 h, while the training time for Fast RCNN with ResNet50 FPN was around 0.93 h. Regarding the inference time, TE-FPN achieved a speed of approximately 11.7 frames/s for images of the same pixel size, whereas FPN achieved a speed of around 13.1 frames/s. As for TIG-DETR as a whole, the training time for the DETR detector with Shifted Window-based Self-Attention was close to 26 h and the inference speed was approximately 3.2 frames/s. After adding our proposed Instance-Based Advanced Guidance Module, the training time increased to about 28 h and the inference speed decreased to around 3.1 frames/s. However, this resulted in an improvement of 1.6 AP in the detection accuracy of large targets without affecting the detection of targets at other scales. In summary, our proposed TIG-DETR achieved a significantly better performance with a slight increase in training and inference time and the overall accuracy of the final model reached 44.1%.

**5. Conclusions**

In this paper, we analyzed the problems of FPNs and DETR-based detectors and found the problems of missing textural information in feature maps, severe confounding effects caused by cross-scale fusion in FPNs, problems of DETR detectors with Shifted Window-based Self-Attention due to the interaction between windows, and the problem of the weakened perception of larger objects by the model due to the limited interaction between windows. Based on these findings, we proposed a new target detection model, TIG-DETR, to further improve the performance of the model. By integrating three simple and effective components, namely enhancing textural information with a bottom-up architecture, Feature-wise Attention, and Instance-Based Advanced Guidance Module, TIG-DETR can significantly improve the baseline approach.

**Author Contributions:** Conceptualization, K.W.; methodology, Z.L. and K.W.; software, Z.L.; validation, Y.W.; formal analysis, C.L.; data curation, K.W.; writing—original draft preparation, Z.L. and K.W.; writing—review and editing, Z.L. and K.W.; visualization, Z.L. and K.W.; supervision, G.L. All authors have read and agreed to the published version of the manuscript.

**Funding:** This research was funded by the Fund of Jilin Provincial Science and Technology Department: Research and development of junior high school physical intelligence experimental platform in mobile environment (20200401087GX).

**Institutional Review Board Statement:** Not applicable.

**Informed Consent Statement:** Not applicable.

**Data Availability Statement:** Not applicable.

**Conflicts of Interest:** The authors declare no conflict of interest.

## Abbreviations

The following abbreviations are used in this manuscript:

| | |
|---|---|
| TE-FPN | Texture-Enhanced FPN |
| TIG-DETR | Texturized Instance Guidance DETR |
| FPN | Three Feature pyramid network |
| SRS | Soft RoI Selection |
| ETA | Enhancing texture information with a bottom-up architecture |
| LFA | Lightweight Feature-wise Attention |
| FWA | Feature-wise Attention |
| IAM | Instance Based Advanced Guidance Module |
| S-IAM | Instance Based Advanced Guidance Module after removal of multiscale fusion |
| W-MSA | Window based Self-Attention |
| SW-MSA | Shift Window based Self-Attention |
| FFN | Feed Forward Networks |
| MLP | Multilayer Perceptron |

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
