# Peer review of "TIG-DETR: Enhancing Texture Preservation and Information Interaction for Target Detection"

_applsci, doi:10.3390/app13148037_

Round 1
Reviewer 1 Report
The paper is an interesting approach in terms of Enhancing Texture Preservation and Information
Interaction for Target Detection. The improved DETR detector utilizes Shifted Window based Self-Attention to replace the multi-headed self-attention module in DETR, thereby accelerating model convergence. Moreover, it incorporates an Instance Based Advanced Guidance Module to enhance instance perception in the image by employing a pre-local self-attentive mechanism for recognizing larger instances. By employing TE-FPN instead of FPN in Faster RCNN with Resnet-50 as the backbone network, we achieve a 1.9% improvement in average accuracy. TIG-DETR achieves an average accuracy of 44.1 with Resnet-50 as the backbone network.
The article is clear, the literature references are sufficient, and the results supported by examples. Experimental results are presented to highlight and validate the proposed approach with support of two case studies.
In a satisfactory manner, the basic purpose of the research has been described, but with some crucial comments that should be taken into consideration.
1. All The final results of the study with the experimental analysis should be written in the abstract including the performance metrics as a comparison with the previous studies.
2. In Figure 2, we advise the authors to summarize the title a suitable one.
3. All the references seem to be old: there is only three reference from 2022 and NO references from 2023. We advise the author to add at least FOUR recent references belonging to 2022 and 2023 to enrich the manuscript.
Author Response
请参阅附件。

Reviewer 2 Report
In this paper, the authors present TIG-DETR, a novel methodology designed to improve texture conservation and information exchange in the context of target detection. The TIG-DETR methodology aims to rectify design inadequacies in both feature pyramid networks (FPN) and transformer-based target detectors, with the potential to enhance the efficacy and precision of target detection systems.
The quality of the writing appears to be clear and concise. The information is presented in a structured manner, providing relevant details. However, there are some concerns regarding the analysis and comparison which are mentioned as follows:
1- The clarification of the contributions in the paper is inadequate and lacks explicit mention in the provided texts.
2- The motivation should be clearly stated in the abstract and introduction.
3- The references are not displayed in order. The authors started with the reference [28].
4- Some acronyms are missing. Some of them are listed later in the manuscript (i.e., SRS (Soft RoI Selection) was provided on page 7. However, the same acronym was cited on page 1.
5- What do the "propositions" in Table 1 refer to? It is recommended to provide specific explanations
6- Please specify the APS, APM, and APL metrics based on Table 1.
7- Please provide more details about the dataset.
8- In line 380, the authors refer to As shown in Table 4 (and not 3). Please confirm.
9- Please provide more details in the discussion section.
10- The conclusion needs to be enhanced. It may be better if the paper's key contribution is highlighted in the Conclusion section.
11- The list of references should be reviewed (please double-check the list again. There are several errors, such as the presence of "//" , [C] and [J]. There is no need to specify if it's a journal paper or a conference. Please follow MDPI Reference List and Citations Style Guide.
Reviewer 3 Report
The presented manuscript is understandable and relevant for the field considered in it. It proposes a new transformer-based approach for detecting objects in images. It is shown that this approach provides an increase in accuracy.
The manuscript is presented in a well-structured form. An overview of approaches to detecting objects in images is presented. The formulation of the research task and the results of its implementation are described. The references given in the review are relevant. The number of references to their work is acceptable. The manuscript contains original ideas. The results obtained are reproducible. The results of the study can be applied to solve practical problems. The illustrative material is designed correctly and reflects the main provisions. The presentation of the materials is made in clear language. The findings are quite consistent with the results of qualitative and quantitative analysis.
No significant drawbacks have been identified. I think that the article can be recommended for publication.
Author Response
Dear Reviewers:
On behalf of all contributors, I would like to sincerely thank you for your letter and for your comments on our article entitled "TIG-DETR: Enhancing Texture Preservation and Information Interaction for Target Detection" (original manuscript number: applsci-2471449) The endorsement. We sincerely thank the reviewers for their enthusiastic work and thank you again for your approval of our article.
Yours sincerely,
Kehan Wang
Corresponding author:
Name: Zhiyong Liu
E-mail: liuzy452@nenu.edu.cn
Reviewer 4 Report
This work offers an interesting and current proposal for improving the efficiency of object detection using advanced feature extraction techniques. The presented issues largely fall within the thematic scope of the journal.
The Authors clearly describe the research need; however, the manuscript requires a few improvements before publication. The experiment shows promising effects of the proposed method, but it is still a few considerations that need to be addressed before the publication of the article:
1. Generally abstract appears to be proper, it provides an overview of the research problem, objectives, methodology, and results of the study. But there are a few things to improve, for example, the abstract could be more clear by avoiding unnecessary repetition of information or jargon that may not be familiar to all readers. Overall, the abstract could benefit from some revisions to better convey the key points of the study in a clear and accessible manner. It is also purposeful to highlight the results achieved during the research.
2. Authors should define the paper's original contribution more clearly with respect to existing literature.
3. It is necessary to add a list of abbreviations due to the presence of a large number of them in the article.
4. How much do the proposed developments affect the time of the object detection process?
5. Line 263 – Mistake in dataset name (“CitysPaces”)
6. Constraints issues related to the proposed method should be expanded (in sections "Results" or "Discussions").
7. According to the reviewer, it is advisable to add a "Discussion" section (or subsection) to the article, which can be partly separated from the "4. Results" section. The purpose of the discussion is to interpret and describe the significance of your findings in light of what was already known about the research problem being investigated, and to explain any new understanding or fresh insights about the problem after you've taken the findings into consideration. The discussion should clearly explain how your study has moved the reader's understanding of the research problem forward from where you left them at the end of the introduction.
Author Response
请参阅附件。
